# Bayesian Optimization on the Cartesian Product of Weighted Graphs to Better Search Discrete Spaces with Irregular Increments

## Abstract

Bayesian optimization is a powerful tool for optimizing a black-box function on a compact Euclidean space under a limited evaluation budget. However, in practice, we may want to optimize over discretization of the solution space. For example, in scientific and engineering problems the discretization of the solution space naturally occurs due to measurement precision or standardized parts. In this work, we consider the problem of optimizing a black-box function with a discretized solution space. To address this problem, prior work uses Bayesian optimization on the Cartesian product of graphs. We extend this work to weighted edges which allow us to exploit the problem structure more effectively. Our proposed method outperforms earlier methods in diverse experiments including neural architecture search benchmarks and physics-based simulations with discretized solution spaces. We also investigate the impact of adding multi-hop edges to weighted graphs, which improves performance of our method on the optimization of synthetic benchmark functions.

## 1 Introduction

Consider a black-box function $f : \mathcal{X} \to \mathbb{R}$ with a compact search space $\mathcal{X} \subset \mathbb{R}^d$. A black-box function has an unknown functional form (Hansen et al., 2010; Turner et al., 2020) and may require many function evaluations in order to determine its optimum. Optimizing black-box functions is thus challenging, particularly when function evaluations are expensive and evaluation budgets are limited. Previous research (Srinivas et al., 2010; Snoek et al., 2012; Turner et al., 2020) has shown that Bayesian optimization (Brochu et al., 2010; Shahriari et al., 2016; Garnett, 2023) is an effective method for optimizing such costly black-box functions. Its effectiveness has also been demonstrated in various real-world applications such as optimization of antireflective glass (Haghanifar et al., 2020), free-electron lasers (Duris et al., 2020), chemical reactions (Shields et al., 2021), and battery lifetimes (Attia et al., 2020).

We consider problems where the original continuous search space must be discretized. In various problem domains, searching for a solution on $\mathcal{X}$ often leads to the repeated evaluation of points that are too close to each other, which is unnecessary and inefficient. This is especially pertinent in engineering problems such as building design, electronic component design, and inventory management. To illustrate, in the context of structural design, choosing structural members, fasteners, materials, connections, and components often demands decisions from a predefined set of standard choices. Likewise, in the design of neural network architectures, diverse variables, i.e., hyperparameters, such as the number of neurons in a layer, learning rate, and output channel size are defined on a discretized search space.

Moreover, considering *irregular increments* such as a logarithmic or geometric sequence, discrete variables are utilized in science and engineering due to physical constraints, fabrication limitations, measurement precision, and ease of interpretability. In particular, this approach is adopted where the range of a variable is very large and the order of magnitude of a variable is more important than its exact value. Some examples include the measurement of earthquake magnitude, sound intensity, and radioactive decay. Electronic components or structural engineering components often come in series where the values follow a logarithmic or geometric

sequence. The need to control various properties and trade-offs in design choices, as well as the ineffectiveness of adjusting variable values by small amounts, also contribute to the use of irregularly discretized variables.

To address the aforementioned practical issue of optimizing a black-box function, we investigate various strategies for optimizing the function on a discretized search space $\mathcal{D} \subset \mathcal{X} \subset \mathbb{R}^d$ using the characteristics of an experiment and the experimenter's design choices. Inspired by the work of Oh et al. (2019), we present a Bayesian optimization method that uses the Cartesian product of weighted graphs. Unlike previous work, our method focuses on a search space of ordinal variables instead of a combinatorial space, where discrete variables with irregular increments are given. Our approach defines a *graph with weighted edges* to represent an ordinal variable. Using the weighted graph, we build a Gaussian process surrogate with diffusion kernels on the Cartesian product of the weighted graphs and maximize an acquisition function on the graph Cartesian product to select the next point. Our algorithm demonstrates improved performance compared to several baseline methods in a range of experiments, including neural network architecture search benchmarks and physics-based simulations. Additionally, we examine the impact of additional multi-hop edges in weighted graphs, and we demonstrate that adding them helps to improve the performance of our method on the optimization of synthetic benchmarks.

We describe the formal problem formulation and our contributions before presenting main concepts.

**Problem Formulation.** A discretized space $\mathcal{D} \subset \mathcal{X}$ of a compact space $\mathcal{X} \subset \mathbb{R}^d$ is defined as

$$\mathcal{D} = \{x_1^{(1)}, \ldots, x_{q_1}^{(1)}\} \times \cdots \times \{x_1^{(d)}, \ldots, x_{q_d}^{(d)}\}, \tag{1}$$

where $x_i^{(k)} < x_j^{(k)}$ if $i < j$ for $k \in [d]$. We define $\mathcal{D}$ as the product of finite sets of candidates for each continuous variable, where the candidates are determined by the experiment's characteristics and the experimenter's design choices; see Section 4.1. Here, we assume that the discretization is relatively coarse, and therefore, the ordinal variables cannot be considered as continuous variables. While this space design process is necessarily handcrafted, it allows us to choose a practically or physically feasible query point based on the experimenter's knowledge, without additional external treatments.

Given that we are optimizing a black-box function, only a function evaluation of a given point $\mathbf{x} \in \mathcal{D}$ is available:

$$y = f(\mathbf{x}) + \epsilon, \tag{2}$$

where $\epsilon$ is observation noise. Therefore, our objective is to find the optimal solution that minimizes the black-box function:

$$\mathbf{x}^\dagger = \arg\min f(\mathbf{x}), \tag{3}$$

where $\mathbf{x} \in \mathcal{D}$.

**Contributions.** Our work can be summarized as follows.

1. We provide an overview of the motivation for search space discretization using irregular increments.

2. We propose a Bayesian optimization strategy for a search space of ordinal variables that is defined on the Cartesian product of weighted graphs.

3. We investigate the effects of introducing additional multi-hop edges in weighted graphs.

4. We demonstrate the superior performance of our approach over existing methods in several experiments, including neural network architecture search benchmarks and physics-based simulations.

We have included our implementation in the supplementary material.

## 2 Related Work

In this section, we discuss prior work related to our paper.

**Bayesian Optimization with Integer-Valued or Ordinal Variables.** Several methods have been proposed to handle integer-valued or ordinal variables in Bayesian optimization. Garrido-Merchán & Hernández-Lobato (2020) analyze two simple methods for integer-valued variables and propose a method with a transformed kernel, which models integer-valued variables directly with a Gaussian process surrogate model. Oh et al. (2019) propose a combinatorial Bayesian optimization method with the Cartesian product of variable-wise graphs, using a chain graph for ordinal variables. Picheny et al. (2019) solve ordinal Bayesian optimization by preserving the ordering of points and using a latent Gaussian process model to determine distances between points. This method is slow, because it requires choosing a large number of free parameters for the Gaussian process.

**Gaussian Processes on Graphs.** Several studies have explored the use of Gaussian processes on graphs. Kondor & Lafferty (2002) propose a diffusion kernel on graphs based on the heat equation, and Smola & Kondor (2003) present a kernel on graphs using the concept of a regularization operator. The diffusion kernel (Kondor & Lafferty, 2002) is a special case of the kernel by Smola & Kondor (2003). Borovitskiy et al. (2021) introduce a Matérn Gaussian process model on graphs, which has several interesting properties including the convergence of a graph Matérn kernel on the Euclidean and Riemannian manifolds. Zhi et al. (2020) propose a spectral kernel learning method for Gaussian processes on graphs, which is capable of learning a spectral kernel on a discrete space. Blanco-Mulero et al. (2021) use a neighborhood kernel on graphs to learn a transition function over time for a dynamic graph structure by measuring the interaction changes between vertices.

**Bayesian Optimization with Prior Knowledge.** By incorporating prior knowledge on the location of solution or information on global optimum, diverse approaches have been proposed. Using the previous similar tasks, approaches to starting an optimization round with better initialization are studied (Poloczek et al., 2016; Lindauer & Hutter, 2018). Moreover, Feurer et al. (2015) propose a method to initialize hyperparameter optimization via meta-learning. Similar to the work (Feurer et al., 2015), an approach to learn meta-features to initialize Bayesian hyperparameter optimization has been suggested (Kim et al., 2017). Also, Perrone et al. (2019) explore a method with the design of data-driven search spaces via transfer learning utilizing historical data. In addition, Ramachandran et al. (2020) investigate the use of priors to warp a search space expanding the space with the prior information, and Souza et al. (2021) propose a method to directly adjust balance between priors and models using the prior information that guides which locations yield better performance. Compared to this line of research, our problem formulation does not employ the prior information on solution locations or global optima, and we consider the measurement precision and standardized parts, which make our problem discrete. As will be discussed in Section 6, we assume that points that are not included in $\mathcal{D}$ cannot be evaluated because they are practically infeasible.

## 3 Background

In order to solve the optimization problem involving discretized continuous variables, introduced in Section 1, we make use of a Bayesian optimization technique. Therefore, we begin by providing a brief description of the Bayesian optimization procedure.

### 3.1 Bayesian Optimization

When optimizing a black-box function using Bayesian optimization, the next point to evaluate is determined sequentially by constructing a surrogate model and maximizing an acquisition function. Because the true function is not explicitly known, a surrogate model is built at each step using the points that have already been evaluated and their corresponding function evaluations. To balance exploration and exploitation in the search space, the surrogate model must provide both a function value estimate and its uncertainty estimate. This can be achieved using a probabilistic regression model, with a Gaussian process (Rasmussen & Williams, 2006) being a popular choice in Bayesian optimization. Once the surrogate model has been constructed, the next query point is selected by maximizing an acquisition function that is based on both the function value and uncertainty estimates produced by the surrogate. These steps are repeated until an evaluation budget is exhausted. For more details, see the work by Brochu et al. (2010); Shahriari et al. (2016); Garnett (2023).

Solving the problem on $\mathcal{D}$ with Bayesian optimization is challenging due to several reasons:

1. A surrogate model on $\mathcal{D}$ should be defined in a distinct manner from a surrogate model for $\mathcal{X}$.

2. Off-the-shelf optimization techniques, such as L-BFGS-B (Liu & Nocedal, 1989) and CMA-ES (Hansen & Ostermeier, 1997), are not suitable for optimizing an acquisition function on $\mathcal{D}$.

3. If we relax the problem from $\mathcal{D}$ to $\mathcal{X}$, it is not trivial to transform back to $\mathcal{D}$.

4. Unlike a combinatorial space with only discrete variables, $\mathcal{D}$ includes ordinal and numerical information that must be considered in the optimization process.

## 3.2 Earlier Methods

**Simple Transformation.** The most basic approach for dealing with integer-valued or ordinal variables in the optimization problem on $\mathcal{D}$ involves solving the problem on a continuous search space $\mathcal{X}$, and then transforming the resulting query point $\mathbf{x}^{\ddagger} \in \mathcal{X}$ to a point in $\mathcal{D}$. Specifically, the closest point in $\mathcal{D}$ to $\mathbf{x}^{\ddagger}$ is selected:

$$\hat{\mathbf{x}}^{\ddagger} = \arg\min_{\mathbf{x} \in \mathcal{D}} \|\mathbf{x} - \mathbf{x}^{\ddagger}\|. \tag{4}$$

Or, to efficiently find the closest point, choose coordinate-wise:

$$\hat{x}_i = \arg\min_{x \in \{x_1^{(i)}, \ldots, x_{q_i}^{(i)}\}} |x - x_i|, \tag{5}$$

for $i \in [d]$ where $\hat{\mathbf{x}}^{\ddagger} = [\hat{x}_1, \ldots, \hat{x}_d]$ and $\mathbf{x}^{\ddagger} = [x_1, \ldots, x_d]$. Henceforth, (4) or (5) is expressed as $\lceil \mathbf{x}^{\ddagger} \rfloor$. However, since this method evaluates a different point than the one chosen by Bayesian optimization, it can be considered as not adhering to the Bayesian optimization policy.

**Continuous Variables Keeping.** This method is similar to the Simple Transformation in that it follows the standard Bayesian optimization process defined on $\mathcal{X}$, and then transforms a query point to a point in $\mathcal{D}$. However, instead of transforming the query point after evaluation, it retains the continuous values of the query points throughout the optimization process. Before the evaluation, the query point $\mathbf{x}^{\ddagger}$ is transformed to $\lceil \mathbf{x}^{\ddagger} \rfloor$ to obtain a response of $\lceil \mathbf{x}^{\ddagger} \rfloor$. While this method aligns with the underlying principles of Bayesian optimization, it can result in the acquisition of points that are rounded to the same integer, thereby leading to suboptimal solutions. The Simple Transformation and the Continuous Variables Keeping methods are thoroughly discussed in (Garrido-Merchán & Hernández-Lobato, 2020).

**Transformed Kernel.** In contrast to the Simple Transformation and Continuous Variables Keeping methods, Garrido-Merchán & Hernández-Lobato (2020) propose a kernel with transformation $k(\lceil \mathbf{x} \rfloor, \lceil \mathbf{x}' \rfloor)$, where $k$ is a kernel for Gaussian process regression, such as the exponentiated quadratic and Matérn kernels, and $\mathbf{x}, \mathbf{x}' \in \mathcal{X}$. The transformed kernel outputs discrete values, making it challenging to optimize an acquisition function with off-the-shelf optimization strategies. Therefore, we sample a sufficient number of candidates from $\mathcal{X}$ to identify the maximizer.

**Chain Graph.** The method proposed by Oh et al. (2019). represents $\mathcal{D}$ as the Cartesian product of graphs, where each graph represents a single ordinal variable. Each variable is modeled as a chain graph, consisting of a vertex matrix $\mathbf{V}_i = [x_1^{(i)}, \ldots, x_{q_i}^{(i)}]$ and an adjacency matrix $\mathbf{A}_i \in \{0, 1\}^{q_i \times q_i}$ for $i \in [d]$, where $\mathbf{A}_i$ is always symmetric and tridiagonial. For example, if $q_i = 3$, $\mathbf{A}_i = \begin{bmatrix} 0 & 1 & 0 \\ 1 & 0 & 1 \\ 0 & 1 & 0 \end{bmatrix}$. The resulting Cartesian product of chain graphs consists of all vertices $\mathbf{V} = \mathbf{V}_1 \times \cdots \times \mathbf{V}_d \in \mathbb{R}^{q \times d}$, where $q = q_1 q_2 \cdots q_d$.

To define a surrogate model on the graph Cartesian product, a diffusion kernel on graphs (Kondor & Lafferty, 2002) is employed, which is computed using the pairs of eigenvalue and eigenvector for the graph Laplacian $\mathbf{L} = \mathbf{D} - \mathbf{A} \in \mathbb{R}^{q \times q}$, where $\mathbf{D}$ and $\mathbf{A}$ are the degree and adjacency matrices, respectively. The diffusion kernel is computed over all vertices $\mathbf{V}$:

$$k(\mathbf{V}, \mathbf{V}) = [\mathbf{v}_1, \ldots, \mathbf{v}_q] \exp(-\beta \mathbf{\Lambda})[\mathbf{v}_1, \ldots, \mathbf{v}_q]^{\top}, \tag{6}$$

(a) Chain Graph

(b) Chain Weighted Graph

Figure 1: Chain graphs without edge weights and with edge weights for a single ordinal variable, and their graph Laplacians. For chain graphs, each edge is called a 1-hop edge.

where $\mathbf{\Lambda} = \mathrm{diag}(\lambda_1, \ldots, \lambda_q)$ is a diagonal matrix of eigenvalues and $\mathbf{v}_1, \ldots, \mathbf{v}_q$ are the respective eigenvectors. See (Kondor & Lafferty, 2002; Seeger, 2004) for a detailed description of the diffusion kernel. To handle large $q$, (6) can be sped up using a Kronecker product decomposition of the Cartesian product of $d$ graphs:

$$k(\mathbf{V}, \mathbf{V}) = \bigotimes_{i=1}^{d} \sum_{j=1}^{q_i} \mathbf{v}_j^{(i)} \exp(-\beta_i \lambda_j^{(i)}) \mathbf{v}_j^{(i)^\top}, \tag{7}$$

where $\bigotimes$ is a Kronecker product and $(\lambda_j^{(i)}, \mathbf{v}_j^{(i)})$ is a $j$th eigenpair of a variable $i$. The work (Oh et al., 2019) provides the details of this decomposition process.

To optimize an acquisition function, a fixed number of query candidates are sampled from $\mathbf{V}$ and a local search is performed, because it is hard to use off-the-shelf optimization tools for optimizing a function over vertices on a graph.

**Probabilistic Reparameterization.** Along with the aforementioned methods, we use the probabilistic reparameterization method proposed by Daulton et al. (2022) as a baseline method. To utilize a gradient-based optimization strategy in optimizing the acquisition function defined on a discrete space or a mixed space of continuous and discrete variables, this method utilizes a technique of probabilistic reparameterization. As reported in the work (Daulton et al., 2022), it shows its effectiveness in diverse circumstances including problems defined on discrete spaces.

## 4 Proposed Method

In this section, we propose a method to optimize a black-box function on a discretized search space.

### 4.1 Search Space Design

In order to optimize a black-box function on $\mathcal{D}$, it is necessary to explicitly define $\mathcal{D} \subset \mathcal{X}$, taking into account not only simple rounding to integers but also precision and irregular increments. As discussed in Section 1, we provide several examples of engineering and science problems such as neural network architecture design and optoelectronic and microfluidic device design. Defining the search space requires careful consideration of the details of the specific experiment, including minimum units for measurement, fabrication, and manufacturing, as well as the experimenter's design choices, such as scaling for infinitesimal or huge values. Further information on search space design can be found in Sections 5 and A.

### 4.2 Weighted Graphs for Ordinal Variables

Here, we present a special cases of weighted graphs for ordinal variables. As will be investigated, we also define a weighted graph with a particular set of multi-hop edges.

---

**Algorithm 1** Bayesian Optimization with W-Graphs

---

**Input:** A time budget $T$ and a black-box function $f$
**Output:** The best solution found $\mathbf{x}^\diamond$
 1: Set up a variable-wise graph structure including edge addition and compute edge weights.
 2: Compute the eigenpairs of the graph Laplacian matrices $\mathbf{L}_1, \ldots, \mathbf{L}_d$ for $d$ weighted graphs.
 3: **for** $t = 1, \ldots, T$ **do**
 4:     Construct a Gaussian process model on the Cartesian product of $d$ weighted graphs with the eigenpairs computed in Line 2.
 5:     Determine the next query point $\mathbf{x}_t^\ddagger$ by maximizing an acquisition function using a local search on the graph Cartesian product.
 6:     Obtain a function value $y_t = f(\mathbf{x}_t^\ddagger) + \epsilon_t$ where $\epsilon_t$ is observation noise.
 7: **end for**
 8: Compute $(\mathbf{x}^\diamond, y^\diamond) = \arg\min_{(\mathbf{x},y) \in \{(\mathbf{x}_i^\ddagger, y_i)\}_{i=1}^T} y$.
 9: **return** $\mathbf{x}^\diamond$

---

We use a chain graph with edge weights to leverage the ordinal and numerical information of a discretized continuous variable. For a variable $i$, we define the edge weight as the distance between two vertices:

$$|x_j^{(i)} - x_k^{(i)}|, \tag{8}$$

for $j, k \in [q_i]$. Using this definition, we define the adjacency matrix for $\mathbf{V}_i$ as follows:

$$[\mathbf{A}_i]_{jk} = \begin{cases} |x_j^{(i)} - x_k^{(i)}| & \text{if } |j - k| = 1, \\ 0 & \text{otherwise,} \end{cases} \tag{9}$$

for $j, k \in [q_i]$. The degree matrix for $\mathbf{V}_i$ is a diagonal matrix with entries $[\mathbf{D}_i]_{jj} = \sum_{k=1}^{q_i} [\mathbf{A}_i]_{jk}$. Using the chain weighted graph, we can compute the graph Laplacian $\mathbf{L} = \mathbf{D} - \mathbf{A}$ and its eigenpairs, which are used to construct a surrogate model.

We can further expand the concept of weighted graphs beyond the chain weighted graphs by including specific edges. As presented in Figures 1 and 5, we refer to an edge between a vertex and $k$-hop vertex as a *k-hop edge* in this paper. For example, edges between $-3.2$ and $1.3$ and between $-3.2$ and $10.1$ are 2-hop and 3-hop edges, respectively. In addition to the graph examples in Figures 1 and 5, we also present their graph Laplacians. Adding extra edges increases the average degree and creates cycles in the graph. However, finding the optimal number and combination of edges is challenging due to a vast number of possible combinations. To address this, we analyze the impact of adding multi-hop edges gradually; see Section 5.4 for the details of these analyses.

### 4.3 Bayesian Optimization with Weighted Graphs

The algorithm we use, as presented in Algorithm 1, follows the similar procedure of standard Bayesian optimization, which is also similar to the framework of the work (Oh et al., 2019). Firstly, we establish a variable-wise graph structure that has its own edges and their corresponding weights. Next, we compute the eigenpairs of the graph Laplacians for $d$ weighted graphs before proceeding with the iterative step to acquire and evaluate a query point. In the iterative step, we construct a Gaussian process surrogate on the graph and maximize an acquisition function to determine the query point.

## 5   Experiments

In this section, we first provide experimental setup for our proposed method as well as the baseline methods explained above. Then, we present our results on three types of experiments including NATS-Bench and physics-based simulations and the impact of additional multi-hop edges.

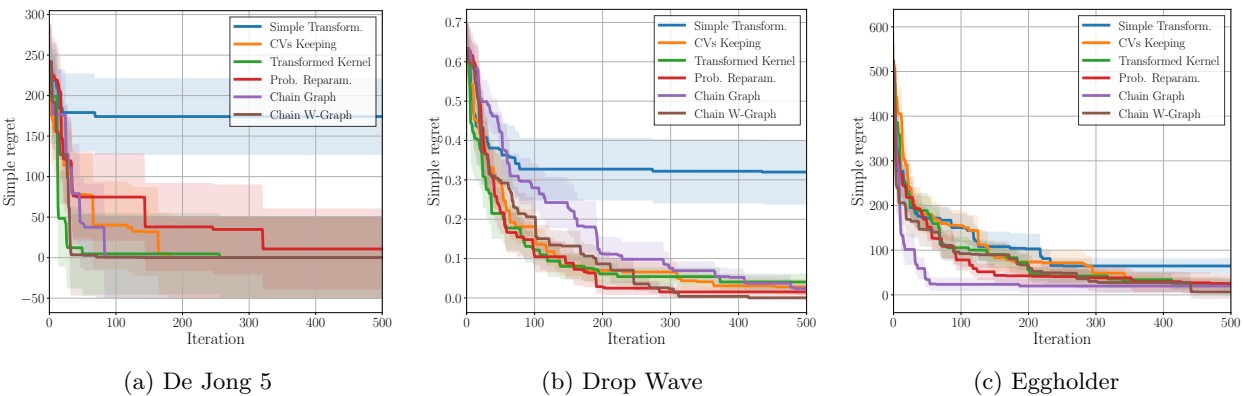

(a) De Jong 5          (b) Drop Wave          (c) Eggholder

Figure 2: Results of optimizing synthetic benchmarks. All experiments are repeated 10 times and the standard error of the sample mean is indicated by the shaded areas. W-Graph and CVs stand for weighted graph and continuous variables. Note that negative shaded regions are a result of statistical representation and not indicative of actual negative regrets.

**Experimental Settings.** For the earlier methods, namely the Simple Transformation, Continuous Variables Keeping, and Transformed Kernel, we use a Gaussian process with the Matérn 5/2 kernel (Rasmussen & Williams, 2006) as the surrogate model. We adopt the expected improvement criterion (Močkus et al., 1978) as the acquisition function for all the methods, including our algorithm. The multi-started L-BFGS-B method is used to optimize the acquisition function in the Simple Transformation and Continuous Variables Keeping methods. In contrast, the Transformed Kernel selects a query point from 10,000 sampled points. For the graph-based approaches, we follow the approach suggested by Oh et al. (2019) and determine the query point by applying a breadth-first local search from the best 20 out of 20,000 randomly sampled vertices. We start all methods with 5 random initial points chosen from the Sobol' sequences (Sobol', 1967), and repeat each experiment 10 times with 10 random seeds, i.e., $42, 84, \ldots, 420$, without any other trials. Our implementation is written in Python with several scientific packages including NumPy (Harris et al., 2020) and SciPy (Virtanen et al., 2020).

## 5.1 Synthetic Functions

We carry out the comparisons of our method and various baselines in synthetic function optimization. To create a discretized search space, we sample a fixed number of points from a compact search space $\mathcal{X}$, and round a query point to the closest point among the points sampled. We uniformly sample 40 points from each variable of $\mathcal{X}$ for a search space design with irregular increments, unless otherwise specified. As shown in Figure 2, our method with chain weighted graphs works better than other existing methods.

## 5.2 NATS-Bench

We tackle a neural network architecture search problem with NATS-Bench (Dong et al., 2021), which provides a testing ground for three popular datasets: CIFAR-10, CIFAR-100, and ImageNet16-120, where each benchmark is controlled by five variables and originally contains 32,768 architecture candidates. To create a search space with irregular increments, we modify the original search space eliminating some of small variable values; see Section A for the details. The intuition of our search space design is that we use fine increments for significant regions and coarse increments for less significant regions by utilizing common knowledge in the deep learning community. As shown in Figure 3, our proposed method with chain weighted graphs finds high-quality solutions faster than other strategies.

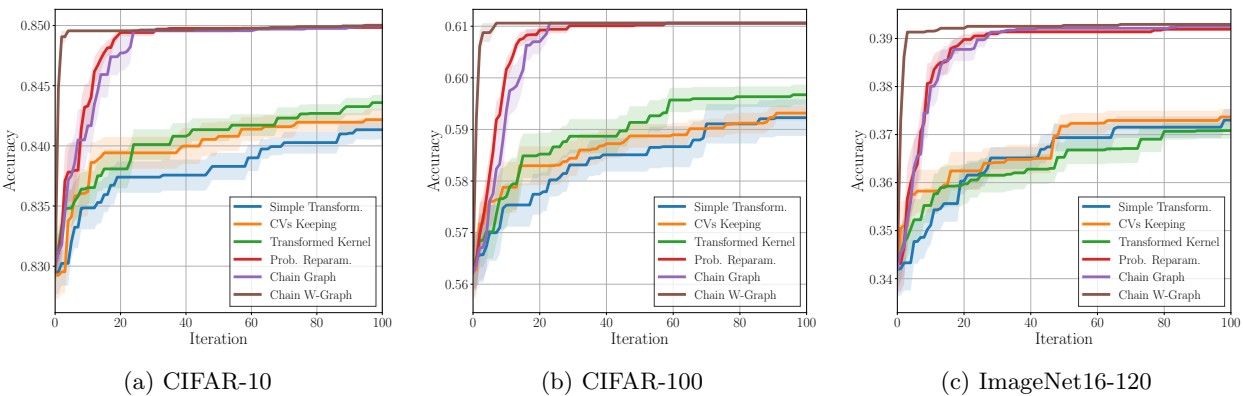

(a) CIFAR-10 (b) CIFAR-100 (c) ImageNet16-120

Figure 3: Bayesian optimization results of NATS-Bench. All experiments are repeated 10 times and the standard error of the sample mean is indicated by the shaded area.

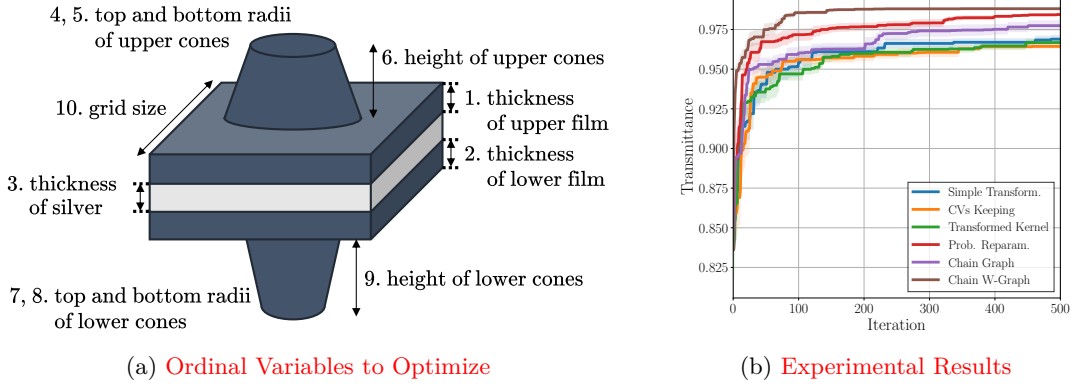

(a) Ordinal Variables to Optimize (b) Experimental Results

Figure 4: Schematic illustration of a structure for electromagnetic shielding and results for optimizing the structure. All experiments are repeated 10 times and the standard error of the sample mean is depicted.

### 5.3 Physics-Based Simulations

We conduct physics-based simulations on electromagnetic shielding as a real-world problem that requires precision in measurement and fabrication. To obtain a response of optical transmission, we assess a nanophotonic structure made of titanium dioxide and silver with the finite difference time-domain method. We utilize Ansys Lumerical software to create and evaluate structures. As discussed in the work of Li et al. (2022), a sandwich structure with double-sided nanocones is effective for transparent electromagnetic shielding. See Figure 4a for a schematic of the structure.

We simulate the transmission at a wavelength of 550 nm by solving the Maxwell's equation in the time domain. Periodic boundary conditions on the sides of the simulation supercell and perfectly matched layers on the top and bottom boundaries of the super cell are applied. We create a mesh grid of minimum size 5 nm over the $x$, $y$, and $z$ directions for upper and lower nanocones and over the $x$ and $y$ directions for upper, lower, and silver film layers. The minimum mesh size over the $z$ direction for each of the three layers is set as 1 nm. We can compute the electromagnetic interference shielding efficiency using the following equation:

$$\text{Shielding Efficiency} = 20 \log \left( 1 + \frac{\eta_0 t_{\text{Ag}}}{2\rho} \right), \tag{10}$$

where $t_{\text{Ag}}$ is the thickness of silver, $\eta_0 = 377 \ \Omega$ is the impedance of free space, and $\rho = 1.59 \times 10^{-8} \ \Omega \cdot \text{m}$ is the bulk resistivity of silver. Since the shielding efficiency (10) depends on the thickness of silver, we choose an optimal structure in terms of transparency. The nanophotonic structure is defined with 10 discretized

Figure 5: Three graph types with $k$-hop edges for a single ordinal variable, and their graph Laplacians

continuous variables. Refer to Figure 4a and Section A for the details of the variables and their ranges. Our method with chain weighted graphs shows the satisfactory performance compared to other methods, as presented in Figure 4b.

## 5.4 Impact of Additional Multi-Hop Edges in Synthetic Benchmarks

To show the impact of additional $k$-hop edges, we evaluate baseline methods and our proposed method on popular synthetic benchmark functions. For this class of problems, continuous variables are discretized by sampling a fixed number of values from a search space $\mathcal{X}$, which makes distances between two adjacent values distinct depending on their sampled values.

The graph Laplacians of weighted graphs with multiple $k$-hop edges are readily computed by following the definition of adjacency and degree matrices. For example, without loss of generality, we can define a complete graph with edge weights, which implies that each vertex is connected to the all other vertices. In this case, an adjacency matrix for a vertex matrix $\mathbf{V}_i$ is defined as the following:

$$[\mathbf{A}_i]_{jk} = \begin{cases} 0 & \text{if } j = k, \\ |x_j^{(i)} - x_k^{(i)}| & \text{otherwise,} \end{cases} \tag{11}$$

for $j, k \in [q_i]$.

Interestingly, as shown in Figure 6, adding extra $k$-hop edges gradually is effective in the performance of Bayesian optimization. For the cases in Figures 6a, 6c, 6d, 6f, 6g, and 6h, the complete weighted graph outperforms the other weighted graphs. This consequence implies that extra edges representing complete distances help in finding a solution. However, as demonstrated in the other cases in Figure 6, the results with the complete weighted graphs are not the best. In particular, in one case for the Branin function, the complete weighted graph shows slower convergence than the other graph types including weighted graphs with 1:2-hop edges, 1:5-hop edges, and 1:10-hop edges. It is worth noting that a weighted graph with 1:$k$-hop edges indicates a weighted graph with 1-hop edges to $k$-hop edges.

To sum up, the use of weighted graphs with 1:$k$-hop edges can improve the connectivity of the graph and consequently, the performance of Bayesian optimization. However, this process of adding edges can be thought of as combinatorial optimization due to the enormous number of possible edge combinations (Ghosh & Boyd, 2006). Thus, we claim that the performance of Bayesian optimization can be improved more by knowing the pertinent structure of a weighted graph, but revealing such structures is left to future work.

Furthermore, while it might seem that a weighted graphs with 1:$k$-hop edges for $k > 1$ are equivalent to weighted graphs with 1-hop edges, i.e., for chain weighted graphs, adding $k$-hop edges for $k > 1$ can increase

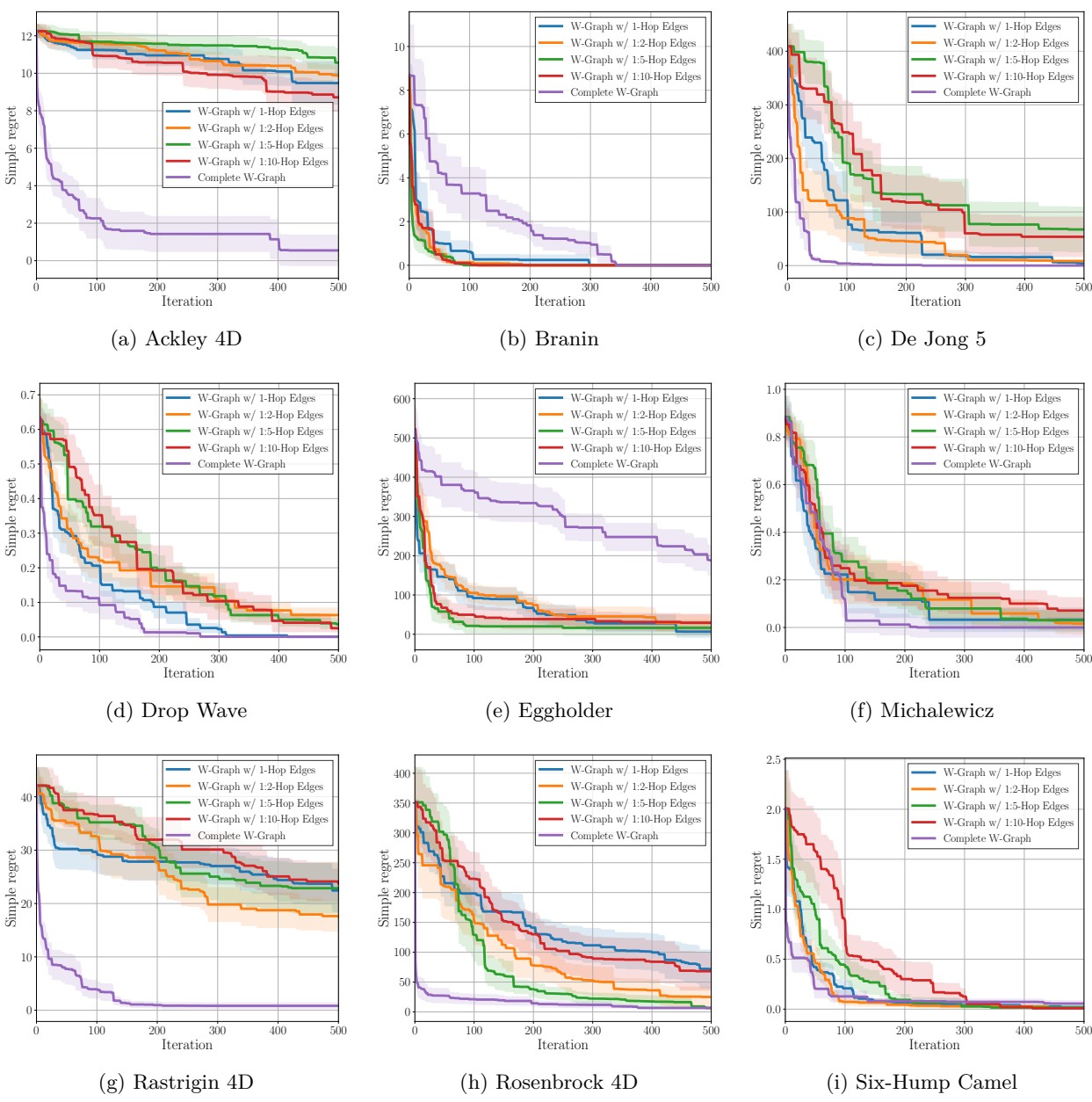

Figure 6: Bayesian optimization results of the effects of adding $k$-hop edges gradually. W-Graph w/ 1:$k$-Hop Edges indicates a weighted graph with $1, \ldots, k$-hop edges. Note that negative shaded regions are a result of statistical representation and not indicative of actual negative regrets.

graph's connectivity. This observation is consistent with the findings in the work (Ghosh & Boyd, 2006), that the Fiedler value (Fiedler, 1973), i.e., the second smallest eigenvalue, increases with increasing average degree, given that all weights are nonnegative and the number of vertices are constant (Holroyd, 2006). Therefore, these edges are not redundant. Moreover, the research by Wainwright et al. (2000); Sudderth et al. (2004), has demonstrated that adding edges and creating cycles in graphs can enhance the expressive power of Gaussian graphical models. These findings suggest that additional edges in the weighted graph can be beneficial for increasing the representation power of the corresponding graph.

## 6    Discussion

In this section, we discuss several topics related to our proposed method and its implications.

**Analysis on Numerical Results by Weighted Graphs.**    Fundamentally, weighted edges help represent the connectivity of variable values and also understand the precise relation between variable values, while unweighted edges only represent their connectivity. Our experimental results thus show that such additional information can improve the performance of Bayesian optimization. However, as can be seen in Figure 6, chain weighted graphs do not always defeat other methods; in addition, complete weighted graphs do not always beat other algorithms. We presume that some edges may be more effective in increasing the expressive power to find a solution, while a few edges may significantly degrade the expressive power. Therefore, the Fiedler value alone may not be sufficient to analyze and interpret Bayesian optimization results. To address this issue, it is necessary to conduct more research on rigorous edge addition and selection in the perspective of Bayesian optimization, and to investigate a representative score for our task.

**Global Solutions in $\mathcal{X} \setminus \mathcal{D}$.**    We assume that points in $\mathcal{X} \setminus \mathcal{D}$ are practically or physically infeasible, making it impossible to evaluate such points; as discussed in our real-world problems in Sections 5.2 and 5.3, there exist points in $\mathcal{X} \setminus \mathcal{D}$ that are impossible to evaluate. As a result, finding a global solution in $\mathcal{X} \setminus \mathcal{D}$ is beyond the scope of this work.

**Limitations.**    While a handcrafted search space may be beneficial in some cases, it can require a great deal of expertise and effort to construct the search space. The ability to systemically identify a practically or physically feasible search space would be valuable in making the method more accessible and widely applicable. It may also help to ensure that the search space is comprehensive and covers all relevant areas, rather than being limited by the expertise or perspective of the experimenter. Future research could explore automated or semi-automated approaches to identifying discrete search spaces.

**Societal Impacts.**    While our work does not have any direct negative societal impacts, it is important to be mindful of any potential ethical implications that may arise in the application of Bayesian optimization in various domains. However, our work can contribute to the advancement of many real-world problems by providing an effective optimization algorithm under certain circumstances, which can ultimately have positive societal impacts.

**Future Directions.**    As previously mentioned, one promising research direction is to develop a technique for delicate edge addition and selection in order to find an optimal graph structure for Bayesian optimization. Another potential future direction for our proposed method with weighted graphs is to explore Bayesian optimization for a space of mixed variables, which is composed of continuous, ordinal, and categorical variables (Daxberger et al., 2020; Oh et al., 2021; Deshwal et al., 2021). Jointly modeling these variables on a graph could be a promising research topic for addressing real-world problems.

## 7    Conclusion

This work addresses practical scientific and engineering problems concerning the precision in measurement and fabrication in the context of Bayesian optimization. In real-world problems, evaluating a query point in continuous space may not be feasible or practical, requiring us to discretize continuous variables based on experiment's characteristics and design choices. To optimize a black-box function on a space of ordinal variables, we explore several approaches and propose an algorithm that leverages the Cartesian product of weighted graphs. We also investigate the impact of multi-hop edges for weighted graphs, and demonstrate that our method outperforms other approaches across diverse experiments including neural network architecture search benchmarks and physics-based simulations.

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

# A    Details of Search Spaces

We present the details of the search spaces used in the experiments.

## A.1    NATS-Bench

Table 1: Search space for NATS-Bench

| Variable | Discrete Variable Values |
|---|---|
| Output channels of 1st convolutional layer | $\{8, 24, 40, 48, 56, 64\}$ |
| Output channels of 1st cell stage | $\{8, 24, 40, 48, 56, 64\}$ |
| Output channels of 1st residual block | $\{8, 24, 40, 48, 56, 64\}$ |
| Output channels of 2nd cell stage | $\{8, 24, 40, 48, 56, 64\}$ |
| Output channels of 2nd residual block | $\{8, 24, 40, 48, 56, 64\}$ |

We use a size search space in NATS-Bench (Dong et al., 2021) as shown in Table 1. To design a search space with irregular increments, we slightly modify the original size search space. More precisely, we eliminate output channel sizes 16 and 32 from each variable, in order to consider the characteristics of the variables; in these experiments large channel sizes are more significant than small channel sizes. As a result, our search space contains 7,776 architecture candidates.

## A.2    Physics-based Simulations

We essentially optimize the following variables:

1. thickness of upper film;

2. thickness of lower film;

3. thickness of silver;

4. top radius of upper cones;

5. bottom radius of upper cones;

6. height of upper cones;

7. top radius of lower cones;

Table 2: Search space for physics-based simulations on electromagnetic shielding. Note that all values except for two ratios are in nanometers.

| Variable | Discrete Variable Values |
|---|---|
| Thickness of upper film | $\{5, 6, 7, 8, 9, 10, 15, 20, \dots, 100\}$ |
| Thickness of lower film | $\{5, 6, 7, 8, 9, 10, 15, 20, \dots, 100\}$ |
| Thickness of silver | $\{3, 4, 5, 10, 15, 20\}$ |
| Ratio of top radius to bottom radius for upper cones | $\{0.01, 0.02, \dots, 0.99\}$ |
| Bottom radius of upper cones | $\{10, 20, \dots, 200\}$ |
| Height of upper cones | $\{50, 60, 70, 80, 90, 100, 150, 200, \dots, 400\}$ |
| Top radius of lower cones | $\{10, 20, \dots, 200\}$ |
| Ratio of bottom radius to top radius for lower cones | $\{0.01, 0.02, \dots, 0.99\}$ |
| Height of lower cones | $\{50, 60, 70, 80, 90, 100, 150, 200, \dots, 400\}$ |
| Grid size | $\{20, 30, \dots, 200\}$ |

8. bottom radius of lower cones;

9. height of lower cones;

10. grid size.

However, to create a physically feasible structure, the structure has to satisfy two constraints that the bottom radius of upper cones is larger than the top radius of upper cones and the top radius of lower cones is larger than the bottom radius of lower cones. Thus, we replace the top radius of upper cones and the bottom radius of lower cones with a ratio of top radius to bottom radius for upper cones and a ratio of bottom radius to top radius for lower cones, respectively. Eventually, we use ordinal variables, which are described in Table 2.

As shown in Table 2, we design a search space for physics-based simulations with irregular increments, i.e., fine increments for relatively small variable values and coarse increments for relatively large variable values, by considering the significance of variables and manufacturing precision.

## B   Impact of Additional Multi-Hop Edges in Real-World Problems

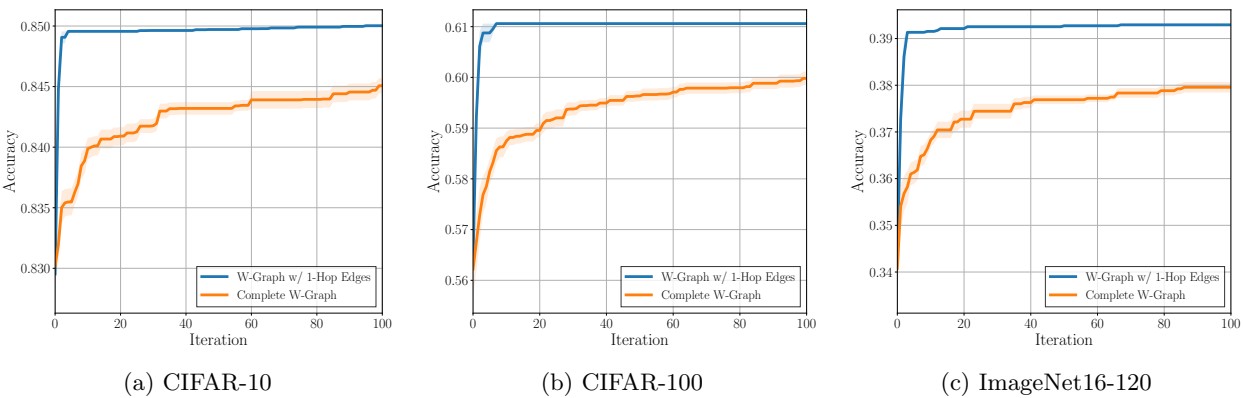

(a) CIFAR-10       (b) CIFAR-100       (c) ImageNet16-120

Figure 7: Results to show the impact of extra multi-hop edges in NATS-Bench

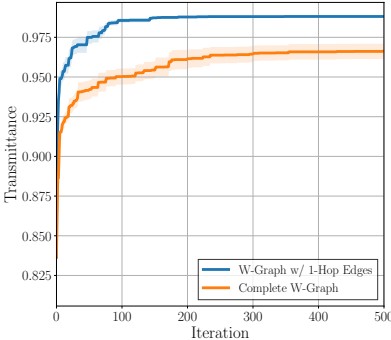

Figure 8: Results to show the impact of extra multi-hop edges in a physics-based simulation for electromagnetic shielding

Similar to the analysis demonstrated in Section 5.4, we present the impact of additional multi-hop edges in real-world problems. Unlike the results in Section 5.4, Bayesian optimization results with chain weighted graphs are better than the results with complete weighted graphs. As discussed in Section 6, we presume that the performance of Bayesian optimization is affected by problem structures, which are practically unknown. Such an interesting analysis on more rigorous edge addition and deletion is left to future work.

## C    Computational Costs for Calculating Eigenvalues and Eigenvectors

Table 3: Additional computational costs for calculating eigenvalues and eigenvectors

| Graph Type | Time (sec.) |
|---|---|
| Chain graph | $0.00841 \pm 0.00724$ |
| Weighted graph with 1-hop edges | $0.01008 \pm 0.00027$ |
| Weighted graph with 1:2-hop edges | $0.01434 \pm 0.00247$ |
| Weighted graph with 1:5-hop edges | $0.01795 \pm 0.00047$ |
| Weighted graph with 1:10-hop edges | $0.02300 \pm 0.00227$ |
| Weighted graph with 1:20-hop edges | $0.03487 \pm 0.00243$ |

We provide elapsed time to compute eigenvalues and eigenvectors in Table 3. To measure the elapsed time, an eight-dimensional synthetic function based on the Ackley function is created where more than 21 variable values exist for each dimension. Also, we conduct this experiment on the same machine by repeating the calculation 1000 times with 10 different random seeds (i.e., 100 times per seed). As expected, adding extra edges leads to more elapsed time. However, we can preemptively compute eigenvalues and eigenvectors at the beginning of a Bayesian optimization round (i.e., Line 2 of Algorithm 1), which implies that it would not be a significant additional burden.

