# OpenReview forum: "Bayesian Optimization on the Cartesian Product of Weighted Graphs to Better Search Discrete Spaces with Irregular Increments"
_TMLR — Rejected by TMLR_

### Review · Reviewer_kZp9 · 2023-06-23

**Summary Of Contributions:**

This paper tackles Bayesian optimization over user-defined coarsely-discretized continuous search spaces. Rather than defining the covariance of the corresponding Gaussian process on this continuous space, the covariance is defined over the Cartesian product of graphs (possibly weighted and with multi-hops). An empirical comparison is provided on several toy examples, a neural network hyperparameter tuning and a physical simulation problems. The addition of the multi-hops is studied as well.

**Audience:**

Yes

**Claims And Evidence:**

Yes

**Requested Changes:**

- [Critical] Improving the motivation of this work in the proposed context and examples.
- [Critical] Considering additional acquisition functions.
-  [Critical] Enhancing the empirical section as suggested above.
- Section 5 arrives too late in the paper, it would be more useful earlier.

Further clarifications:
By taking into account multi-hops, is it akin to reproduce the original inter distances in the compact search space?


**Strengths And Weaknesses:**

# Strengths
- The use of graph-based Gaussian processes for Bayesian optimization is much less studied than their continuous counterparts.
- The study of the effect of multi-hops is interesting.

# Weaknesses
First of all, the use of a graph structure on a design space that is initially a compact set seems contrived. It could make sense on other problems with categorical variables, or on complex objects like molecules, see, e.g., Korovina, K., Xu, S., Kandasamy, K., Neiswanger, W., Poczos, B., Schneider, J., & Xing, E. (2020, June). Chembo: Bayesian optimization of small organic molecules with synthesizable recommendations. In International Conference on Artificial Intelligence and Statistics (pp. 3393-3403). PMLR.

Indeed, there are ordinal variables in neural networks, but the proposed setup relates more to the inclusion of prior knowledge, e.g., Souza, A., Nardi, L., Oliveira, L. B., Olukotun, K., Lindauer, M., & Hutter, F. (2021). Bayesian optimization with a prior for the optimum. In Machine Learning and Knowledge Discovery in Databases. Research Track: European Conference, ECML PKDD 2021, Bilbao, Spain, September 13–17, 2021, Proceedings, Part III 21 (pp. 265-296). Springer International Publishing.
For the electromagnetic shielding test case, the simulator would run with continuous variables.

Then the choice of the acquisition function is not discussed, and possibly not the best fit for the proposed setup. For instance, it could be an entropy defined on the discrete search space, Thompson sampling or conditional improvement (discussed, e.g., in Gramacy, R. B. (2020). Surrogates: Gaussian process modeling, design, and optimization for the applied sciences. CRC press.). For computer experiments, being able to evaluate everywhere can bring more information to improve the surrogate compared to limiting the search to a few locations.
As a side remark, using coarse discrete sets for acquisition function search is a standard simplification of the continuous problem.

There are missing details in the empirical section:
- what is the metric used? For synthetic test functions, if the search space is random, so is the value of the optimum. Using the regret would be preferred in this context.
- what is the search space used by the competitors (e.g., continuous, discrete)?
- some methods use 10k sampled points for acquisition function search, another 20k plus local search, how is this chosen?

Finally, the empirical comparison could include state of the art methods, say: Eriksson, D., Pearce, M., Gardner, J., Turner, R. D., & Poloczek, M. (2019). Scalable global optimization via local bayesian optimization. Advances in neural information processing systems, 32.
or Cowen-Rivers, A. I., Lyu, W., Tutunov, R., Wang, Z., Grosnit, A., Griffiths, R. R., ... & Bou-Ammar, H. (2022). HEBO: pushing the limits of sample-efficient hyper-parameter optimisation. Journal of Artificial Intelligence Research, 74, 1269-1349.

---

> ### Author Response · Authors · 2023-07-05
> **Response to Reviewer kZp9 (1/2)**
>
> We sincerely appreciate the Reviewer's comments and constructive feedback.  We are encouraged by the recognition of the novel aspects of our approach, such as the utilization of graph-based Gaussian processes for Bayesian optimization and the study of multi-hops.  We have made the following changes to the paper to enhance its quality:
>
> > First of all, the use of a graph structure on a design space that is initially a compact set seems contrived. It could make sense on other problems with categorical variables, or on complex objects like molecules, see, e.g., Korovina, K., Xu, S., Kandasamy, K., Neiswanger, W., Poczos, B., Schneider, J., & Xing, E. (2020, June). Chembo: Bayesian optimization of small organic molecules with synthesizable recommendations. In International Conference on Artificial Intelligence and Statistics (pp. 3393-3403). PMLR.
>
> We think that ChemBO is not suitable for our problem formulation.  While an object like molecules is defined on a search space over graphs (or between graphs), a solution of our interest is defined on a search space on a graph, which is expressed by the Cartesian product of sub-graphs in our paper.  Therefore, our problem on a graph is discrete -- by generalizing this problem, the problem becomes combinatorial.
>
> > Indeed, there are ordinal variables in neural networks, but the proposed setup relates more to the inclusion of prior knowledge, e.g., Souza, A., Nardi, L., Oliveira, L. B., Olukotun, K., Lindauer, M., & Hutter, F. (2021). Bayesian optimization with a prior for the optimum. In Machine Learning and Knowledge Discovery in Databases. Research Track: European Conference, ECML PKDD 2021, Bilbao, Spain, September 13–17, 2021, Proceedings, Part III 21 (pp. 265-296). Springer International Publishing.
>
> Thank you for recommending the related work.  We have added discussion on the research you mentioned in *Section 2*; please see *the paragraph "Bayesian Optimization with Prior Knowledge"* in *Section 2*.  We contrast previous work and that in our paper:
>
> `Compared to this line of research, our problem formulation does not employ the prior information on solution locations or global optima, and we consider the measurement precision and standardized parts, which make our problem discrete. As will be discussed in Section 6, we assume that points that are not included in $\mathcal{D}$ cannot be evaluated because they are practically infeasible.`
>
> > For the electromagnetic shielding test case, the simulator would run with continuous variables.
>
> In electrodynamic simulations, the finite difference method discretizes space and time to solve Maxwell's equations.  Therefore, the discretization of continuous variables is natural application of our method.  As many real world engineering and science problems (such as the building design, electronic component design, and inventory management) involve discrete spaces or choices, this electromagnetic shielding test case helps serve as a motivating example.
>
> > Then the choice of the acquisition function is not discussed, and possibly not the best fit for the proposed setup.
>
> As reported in the previous work [3, 4], the expected improvement, which is used in our paper, is one of popular choices for an acquisition function.  Since we used the expected improvement as an acquisition function for all methods, we think that it is fair enough to compare our methods to existing methods.  As mentioned later, we added an extra baseline method [1] in order to enhance the experimental part.
>
> > what is the metric used? For synthetic test functions, if the search space is random, so is the value of the optimum. Using the regret would be preferred in this context.
>
> Thank you for pointing this out.  We have updated the text and figures to report regrets instead of function values; please see *Section 5* of the revision.
>
> > what is the search space used by the competitors (e.g., continuous, discrete)?
>
> In this paper, we only consider a discrete space as a search space.  Thus, all search spaces for the competitors are discrete.  Discussion on this issue is described in *Section 6*.
>
> > some methods use 10k sampled points for acquisition function search, another 20k plus local search, how is this chosen?
>
> We followed the settings of COMBO for a fair comparison.  For the problem sizes solved in this paper, 10k and 20k showed similar performance in the preliminary experiments on synthetic functions.
>
> > Finally, the empirical comparison could include state of the art methods.
>
> We have included a new baseline Bayesian optimization method for problems defined on a discrete search space [1] for comparison.  This method has been shown to be superior to various methods such as HyBO and a modified version of Casmopolitan [2].  As shown in *Section 5* of the revision, our method outperforms the existing methods including Probabilistic Reparameterization [1].

---

> ### Author Response · Authors · 2023-07-05
> **Response to Reviewer kZp9 (2/2)**
>
> > Section 5 arrives too late in the paper, it would be more useful earlier.
>
> We have reorganized the paper to improve its structure and presentation.  We have moved *Section 5* to the beginning of the paper.
>
> > Further clarifications: By taking into account multi-hops, is it akin to reproduce the original inter distances in the compact search space?
>
> Yes, it is akin to reproduce the original distances in the Euclidean space.  As described above, we would like to emphasize that our interest is to tackle Bayesian optimization on a discrete space using the Cartesian product of graphs.
>
> [1] Samuel Daulton, et al. Bayesian Optimization over Discrete and Mixed Spaces via Probabilistic Reparameterization. Advances in Neural Information Processing Systems, 2022.
>
> [2] Xingchen Wan, et al. Think Global and Act Local: Bayesian Optimisation over High-Dimensional Categorical and Mixed Search Spaces. International Conference on Machine Learning, 2021.
>
> [3] Peter I. Frazier. A tutorial on Bayesian Optimization. arXiv preprint arXiv:1807.02811, 2018.
>
> [4] Roman Garnett. Bayesian Optimization. Cambridge University Press, 2023.

---

### Review · Reviewer_CAVv · 2023-06-25

**Summary Of Contributions:**

This paper proposes a Bayesian optimization method for scenarios where all tunable parameters are discrete and ordinal. To tackle this problem, the paper represents ordinal parameters as weighted graphs and use a GP with a diffusion kernel. This builds off of COMBO (Oh et al., 2019). The paper empirically evaluates this method on several benchmarks including neural architecture search and physics-based problems.

**Audience:**

Yes

**Claims And Evidence:**

Yes

**Requested Changes:**

Add comparisons with baselines and complete-w graph results for main experiments.

**Strengths And Weaknesses:**

## Strengths
The idea is intuitive and appealing and the empirical results are compelling.

The paper is largely an incremental improvement upon COMBO (using weighted graphs to represent ordinals). Nevertheless the performance improvements are strong, and I think the community will find this work to be of interest.

## Weaknesses

[minor] The empirical results are a bit perplexing. In the real-world experiments, the Chain-w graph method performs best. But in the experiments on synthetic benchmarks the Complete-w graph before much better than W-graph w/ 1-hop edges---which is the chain-w graph based on my understanding. Given these results (Figure 6), why is the complete-w graph not used in the real-world experiments in Figures 3 and 4?

[minor] The paper does not compare against recent works on Bayesian optimization with ordinal parameters. Such as Casmopolitan (Wan et al, 2021) and probabilistic reparameterization (Daulton et al, 2022).

---

> ### Author Response · Authors · 2023-07-05
> **Resposne to Reviewer CAVv**
>
> We appreciate the Reviewer's thoughtful review and positive comments on the intuitive appeal and compelling empirical results of our paper.  We are encouraged by the acknowledgement of our performance improvements and have made the following changes to paper to address concerns raised:
>
> > [minor] The empirical results are a bit perplexing. In the real-world experiments, the Chain-w graph method performs best. But in the experiments on synthetic benchmarks the Complete-w graph before much better than W-graph w/ 1-hop edges---which is the chain-w graph based on my understanding. Given these results (Figure 6), why is the complete-w graph not used in the real-world experiments in Figures 3 and 4?
>
> We have added the results of the complete weighted graph for the real-world experiments in the appendix of the revision.
>
> For your point on the experiments, the complete weighted graph is overly flexible and it might interfere with a surrogate model depending on problem structures.  As described in *Section 6*, some edges may be more effective in increasing the expressive power to find a solution, while a few edges may significantly degrade the expressive power.  Moreover, the Fiedler value, which is the second smallest eigenvalue, is not sufficient to analyze our Bayesian optimization results.  To investigate this issue more, it is necessary to conduct research on rigorous edge addition and selection in the perspective of Bayesian optimization.  This is an interesting topic for future work.
>
> > [minor] The paper does not compare against recent works on Bayesian optimization with ordinal parameters. Such as Casmopolitan (Wan et al, 2021) and probabilistic reparameterization (Daulton et al, 2022).
>
> We have included a new baseline Bayesian optimization method for problems defined on a discrete search space [1] for comparison.  This method has been shown to be superior to various methods such as HyBO and a modified version of Casmopolitan [2].  According to [https://github.com/xingchenwan/Casmopolitan](https://github.com/xingchenwan/Casmopolitan), Casmopolitan [2] is a method for categorical variables and mixed spaces of categorical/continuous variables and its direct extension to discrete variables is the method proposed by Daulton et al. [1].  As shown in *Section 5* of the revision, our method outperforms the existing methods including Probabilistic Reparameterization [1].
>
> [1] Samuel Daulton, et al. Bayesian Optimization over Discrete and Mixed Spaces via Probabilistic Reparameterization. Advances in Neural Information Processing Systems, 2022.
>
> [2] Xingchen Wan, et al. Think Global and Act Local: Bayesian Optimisation over High-Dimensional Categorical and Mixed Search Spaces. International Conference on Machine Learning, 2021.

---

> > ### Author Response · Authors · 2023-07-15
> > **Resposne to Reviewer CAVv**
> >
> > We appreciate your constructive feedback again.  As described in our previous response, we tried to answer your questions and resolve your concerns.  Please let us know if there are additional questions and concerns.

---

### Review · Reviewer_uB17 · 2023-07-02

**Summary Of Contributions:**

The paper extends the previous Bayesian Optimization (BO) work that uses Catersian product of graphs [Oh et al, 2019] to allows BO to work with discrete search space. The main idea of the paper is to use weighted edges where the weights are computed by the distance between the two vertices of the graph. The BO procedure is then conducted as in Oh et al, 2019 to find the global optimum of the optimization problem.

**Audience:**

Yes

**Claims And Evidence:**

No

**Requested Changes:**

1. More insight on why the proposed weighted Cartesian graph works.
2. Experiments need to include more recent BO methods that tackle the problem of discrete search space (please find the references in the previous section).
3. Experiments need to explain in more detail the dimensions of the problems used, and also should include more high-dimensional problems.


**Strengths And Weaknesses:**

Strengths:
+ The paper’s writing is generally clear. The problem setting, some previous related methods are described clearly. The experiments are explained in detail with ablation study being conducted to understand some aspects of the proposed technique.
+ The idea of using weighted Cartesian graphs for BO seems to be interesting and might be worth for further investigation.

Weaknesses
- The paper doesn’t provide much insight why it makes sense to use weighted Cartesian graph with the weights defined by the distances between the vertices. I don’t understand why the technique of setting the weights by the distance between the vertices will work.
- The experiments seem to lack of the comparison with some new methods that tackle the problem of BO with discrete search space. For example, the work “Bayesian Optimization over Hybrid Spaces” by Deshwal et al (ICML 2021), and the work “Bayesian Optimization over Discrete and Mixed Spaces via Probabilistic Reparameterization” by Daulton et al (NeurIPS 2022).
- The dimensions of the optimization problems used in the paper need to be described in more detail, especially the NATS-Bench. Besides, it seems to me the dimensions of the optimization problems used in this paper are quite small (at least I know that for synthetic problems, the dimensions are just up to 4, and it’s unclear on the dimensions of the NATS-Bench problems).
- In Section 4.1, it says that 40 data points are sampled from X to construct the discrete search space. Why only 40? This seems to be quite small.

---

> ### Author Response · Authors · 2023-07-05
> **Response to Reviewer uB17**
>
> We appreciate the Reviewer's valuable insights and constructive comments.  We thank the Reviewer for acknowledging the clarity of the paper's writing and detail in the experiments as strengths.  We have modified the manuscript based on the Reviewer's comments as follows:
>
> > More insight on why the proposed weighted Cartesian graph works.
>
> We have included a more detailed explanation on why our method works well in *the paragraph "Analysis on Numerical Results by Weighted Graphs"* in *Section 6*.
>
> Weighted edges help represent the connectivity of variable values and also understand the precise relation between variable values, while unweighted edges only represent their connectivity.  Our experimental results thus show that such additional information can improve the performance of Bayesian optimization.  However, the complete weighted graph is overly flexible and it might interfere with a surrogate model depending on problem structures.  As described in *Section 6*, some edges may be more effective in increasing the expressive power to find a solution, while a few edges may significantly degrade the expressive power.  Moreover, the Fiedler value, which is the second smallest eigenvalue, is not sufficient to analyze our Bayesian optimization results.  To investigate this issue more, it is necessary to conduct research on rigorous edge addition and selection in the perspective of Bayesian optimization.  This is an interesting topic for future work.
>
> > Experiments need to include more recent BO methods that tackle the problem of discrete search space (please find the references in the previous section).
>
> We have included a new baseline Bayesian optimization method for problems defined on a discrete search space [1] for comparison.  This method has been shown to be superior to various methods such as HyBO and a modified version of Casmopolitan [2].  As shown in *Section 5* of the revision, our method outperforms the existing methods including Probabilistic Reparameterization [1].
>
> > The dimensions of the optimization problems used in the paper need to be described in more detail, especially the NATS-Bench. Besides, it seems to me the dimensions of the optimization problems used in this paper are quite small (at least I know that for synthetic problems, the dimensions are just up to 4, and it’s unclear on the dimensions of the NATS-Bench problems).
>
> Thank you for pointing the need to provide detailed information regarding the dimensions of the optimization problems used in the experiments.  The revised paper has been updated with the required details.  The dimensionality of the NATS-Bench problem is 5.  Our physics-based simulation problem has 10 variables and $37 \ 069 \ 796 \ 966 \ 400$ solution candidates, so that it should be considered a large-scale problem.
>
> > In Section 4.1, it says that 40 data points are sampled from X to construct the discrete search space. Why only 40? This seems to be quite small.
>
> We acknowledge the need to clarify the choice of data points. We sampled 40 points from each variable in $\mathcal{X}$.  For four-dimensional problems, there exist $40^4 = 2 \ 560 \ 000$ solution candidates.  The text has been updated accordingly.
>
> [1] Samuel Daulton, et al. Bayesian Optimization over Discrete and Mixed Spaces via Probabilistic Reparameterization. Advances in Neural Information Processing Systems, 2022.
>
> [2] Xingchen Wan, et al. Think Global and Act Local: Bayesian Optimisation over High-Dimensional Categorical and Mixed Search Spaces. International Conference on Machine Learning, 2021.

---

> > ### Author Response · Authors · 2023-07-15
> > **Response to Reviewer uB17**
> >
> > We appreciate your constructive feedback again.  As described in our previous response, we tried to answer your questions and resolve your concerns.  Please let us know if there are additional questions and concerns.

---

> > > ### Comment · Reviewer_uB17 · 2023-07-19
> > > **Reply to the authors' response**
> > >
> > > Dear authors,
> > >
> > > Thank you for your response and sorry for the late reply.
> > >
> > > The response has addressed my 2nd and 3rd question, however, it hasn't cleared my first question. In my first question, I asked for some insights on why the proposed weighted Cartesian graph works. In particular, I don't understand why setting the adjacent matrix using the formula in Eq. (9) makes the proposed method to perform well. Why don't we set using other ways? Are there any theoretical insights about the choices of these settings (can be from previous published works). The new paragraph added in Section 6 just give some very general comments about the weighted graph in general (no relation to the formula used in Eq. (9)) and mentions about the numerical results. I expect more insight regarding the choice of Eq. (9) to construct the weighted graph.

---

> > > > ### Author Response · Authors · 2023-07-19
> > > > **Response to Reviewer uB17**
> > > >
> > > > We are glad that the 2nd and 3rd questions have been resolved.
> > > >
> > > > For the first question, we would like to specify our answer more.
> > > >
> > > > Our surrogate model (i.e., Gaussian processes on graphs) is built with a diffusion kernel.  The similarity between nodes on a graph, which is computed by the kernel, plays an important role for constructing the surrogate model that appropriately represents relationship between nodes, where the similarity between nodes is determined by two factors: *connectivity* and *distance between nodes*.  Unlike the previous work that only considers the connectivity, our formulation with Eq. (9) encourages the surrogate model to utilize both crucial factors.
> > > >
> > > > The importance of both factors has been discussed in the research on *spectral clustering*.  The use of distances between nodes (or data points in this problem) is essential for finding clusters on a graph representation.  Without the distance information, it is hard to determine groups of data points on the Euclidean space.  Likewise, the gain of our proposed method is also led by utilizing both factors.
> > > >
> > > > *We are going to revise our submission accordingly.*
> > > >
> > > > Furthermore, we agree with your point that we do not need to limit the form of Eq. (9).  We can set it up as other forms like $d^2$ or $\sqrt{d}$ where $d$ is the distance between two nodes.  It will be left to future work.

---

> > > > > ### Comment · Reviewer_uB17 · 2023-07-22
> > > > > **Thank you for your response**
> > > > >
> > > > > I feel clearer regarding the motivation of using distance in the adjacent matrix now. The authors should explain about this in more detail in the paper, otherwise, it's really hard to understand why the adjacent matrix for the weighted graph is set using the distance information.

---

> > > > > > ### Author Response · Authors · 2023-07-25
> > > > > > **Response to Reviewer uB17**
> > > > > >
> > > > > > We are happy that it becomes clearer.  We will revise the final version of our submission accordingly.

---

### Author Response · Authors · 2023-07-05
**Changes in the revision**

We thank the Reviewers for their generally positive feedback and their feedback that our work is of interest to the TMLR audience.  In particular, Reviewer uB17 commented that "***The idea of using weighted Cartesian graphs for BO seems to be interesting***", Reviewer CAVv mentioned that "***The idea is intuitive and appealing and the empirical results are compelling***" and "***The performance improvements are strong, and I think the community will find this work to be of interest***", and Reviwerer kZp9 commented that "***The use of graph-based Gaussian processes for Bayesian optimization is much less studied than their continuous counterparts***" and "***The study of the effect of multi-hops is interesting.***"

Based on the Reviewer's comments and suggestions, we have made the following improvements to our paper:
* Added Probabilistic Reparameterization [1] as a baseline
* Changed metrics for synthetic functions to simple regrets
* Moved *the section "Related Work"* to *Section 2*
* Added *the paragraph "Bayesian Optimization with Prior Knowledge"* in *Section 2*
* Enhanced *the paragraph "Analysis on Numerical Results by Weighted Graphs"* in *Section 6*
* Added *the section "Computational Costs for Calculating Eigenvalues and Eigenvectors"*
* Revised minor issues

[1] Samuel Daulton, et al. Bayesian Optimization over Discrete and Mixed Spaces via Probabilistic Reparameterization. Advances in Neural Information Processing Systems, 2022.

---

### Decision · Action_Editors · 2023-08-11

**Recommendation:** Reject

**Comment:**

The work is closely based on previous work by Oh et al, essential extending their (unweighted) graph approach to weighted graphs. It further includes investigations into the role of the number of hops to include.

The work is interesting but feels preliminary. I largely agree with the majority of the reviewers that further work would be needed to

- better motivate the work,
- investigate the impact of the acquisition function optimisation,
- provide theoretical support for the weighted chain graphs, and,
- in light of figures 6-8, tie up the lose ends with the number of hops to include (and hence which method/graph to use).

**Audience:**

The work will be of interest to readers of TMLR. However, the motivation for the particular problem setup needed to be improved. While the motivation became clearer during the discussion, I think the paper still makes the readers work too hard to understand the motivation. The paper would here benefit from a revision, taking the finer points of the discussion with Rev kZp9 fully into account.

**Claims And Evidence:**

The paper proposes a set of new methods for Bayesian optimisation over discrete spaces that are based on previous work by Oh et al, NeurIPS 2019. While the reviewers appreciated the paper, they flagged missing theoretical support for the approach and pointed out the large variability in the results obtained by the proposed methods (complete graph vs chain weighted graph with different number of hops), leaving the reader wonder what method to choose and why there is such a large variability.

**Resubmission Of Major Revision:**

The authors may consider submitting a major revision at a later time.